# Design and Preparation of Flexible Graphene/Nonwoven Composites with Simultaneous Broadband Absorption and Stable Properties

**DOI:** 10.3390/nano13040634

**Published:** 2023-02-05

**Authors:** Song Bi, Yongzhi Song, Genliang Hou, Hao Li, Nengjun Yang, Zhaohui Liu

**Affiliations:** 1304 Department, Xi’an Research Institute of High-Tech, Xi’an 710025, China; 2College of Weapon Science and Technology, Xi’an Technological University, Xi’an 710025, China

**Keywords:** RGO/NW, Jaumann absorber, performance stability, microwave absorption, reflection loss

## Abstract

As the world moves into the 21st century, the complex electromagnetic wave environment is receiving widespread attention due to its impact on human health, suggesting the critical importance of wearable absorbing materials. In this paper, graphene nonwoven (RGO/NW) composites were prepared by diffusely distributing graphene sheets in a polypropylene three-dimensional framework through Hummers’ method. Moreover, based on the Jaumann structural material design concept, the RGO/NW composite was designed as a multilayer microwave absorber, with self-recovery capability. It achieves effective absorption (reflection loss of −10 dB) in the 2~18 GHz electromagnetic wave frequency domain, exhibiting a larger bandwidth than that reported in the literature for absorbers of equivalent thickness. In addition, the rationally designed three-layer sample has an electromagnetic wave absorption of over 97% (reflection loss of −15 dB) of the bandwidth over 14 GHz. In addition, due to the physical and chemical stability of graphene and the deformation recovery ability of nonwoven fabric, the absorber also shows good deformation recovery ability and stable absorption performance. This broadband absorption and extreme environmental adaptability make this flexible absorber promising for various applications, especially for personnel wearable devices.

## 1. Introduction

With the development of science and technology, electronic information is more and more widely used in daily life, and with its use comes a more hostile electromagnetic environment. Long-term exposure to complex electromagnetic environments is bound to affect people’s health, so it is essential to study lightweight and flexible broadband absorbent materials suitable for personnel to wear. Graphene, a new type of carbon nanomaterial, has shown unique advantages in electromagnetic wave absorption due to its low density and superior physical and chemical properties [1,2]. In particular, research on the construction of 3D graphene has received increasing attention because it not only maintains the intrinsic properties of graphene, but also exhibits more advanced properties by virtue of its structure [3,4,5]. Compared to 2D graphene sheets, 3D graphene can be more easily tuned in terms of conductivity and physical structure by adjusting the initial material and reduction conditions. Furthermore, it also overcomes the severe aggregation problem of graphene layers during material preparation [6,7]. It has been reported that 3D graphene-structured absorbers designed with appropriate chemical composition and physical structure can achieve a relatively perfect balance between good impedance matching and high loss, resulting in better electromagnetic wave absorption performance [8,9,10].

The widely studied graphene aerogel is a typical 3D graphene material, a porous foam-like carbon material created by an autonomous assembly of templates or non-templates [11,12]. The reduced graphene oxide/nonwoven fabric (RGO/NW) composite studied in this work, as a carbon/polymer composite, is similar to a porous graphene aerogel composite prepared by distributing graphene in a template of a nonwoven matrix [13]. Since the polypropylene resin fibers used to prepare the nonwoven material are electromagnetic wave lossless materials, such as air, the RGO/NW composite has similar wave absorption properties to aerogels under the same carbon filling amount and microstructure. The difference is that the density of the RGO/NW composite is slightly higher than that of the foam material [14]. However, it can be prepared at a lower cost with better structural and absorption stability, which makes it suitable for electromagnetic wave absorption and protection in extremely harsh environments.

The excellent absorption properties of electromagnetic wave absorbing materials require extremely complex and optimized design solutions and preparation processes, placing greater demands on the performance of their constituent materials. A layered structure design can meet the performance requirements by varying each layer’s thickness and dielectric properties, superimposing them. The most representative design methodology is the Jaumann absorber [15,16,17], which improves the absorbing bandwidth by increasing the number of resistive sheets and spacers. When the conductive polymer matrix composites were gradually applied in the field of wave absorption, it was found that by replacing resistive sheets and spacers with conductive polymer matrix composites with different electromagnetic properties, a multilayer wave absorption material, with suitable impedance and excellent wave absorption performance, could be obtained through an optimized combination. It is characterized by using the composite dielectric layers of recursive superposition to meet the dielectric layer impedance progressive change, while enhancing the loss characteristics of electromagnetic waves to achieve the resonant absorption of structural absorbers in a wide frequency domain.

In this work, graphene oxide (GO) suspensions were prepared by the Hummers’ method using a nonwoven fabric as the structural framework, and subsequently dispersed uniformly in the gaps of the nonwoven fabric fibers before being made into graphene aerogel/nonwoven fabric bi-3D structural composites by in situ reductions. The concentration of graphene in the RGO/NW composites was adjusted by the concentration of GO (3, 5, 7 and 9 mg/mL) during the preparation process. The effect of graphene concentration on the dielectric properties of the composites was investigated, and their electromagnetic wave response behaviors were summarized. Based on the design concept of Jaumann absorber, the superposition method of gradient concentration media was used to prepare and study the wave absorbing properties of multilayer carbon-based nonwoven composites. Finally, a series of environmental adaptation tests are conducted to verify the environmental adaptability and stability of the RGO/NW composites.

## 2. Experiments

### 2.1. Preparation of RGO/NW Composites

Since graphene is usually insoluble in water and tends to agglomerate in water, it is difficult to disperse graphene directly in the nonwoven fabric to form an ideal RGO/NW composite. In this paper, reduced graphene oxide/nonwoven fabric composites were prepared by the Hummers’ method based on graphene oxide, as shown in Figure 1. In the preparation of graphene oxide (GO) by the Hummers’ method, negatively charged functional groups with phenolic groups and carboxyl groups are formed on the surface of GO to inhibit GO agglomeration so that GO can be dispersed relatively uniformly in water. The hydrophilic nature of GO means that water molecules can enter between the layers of GO more easily, widening the layer spacing to 0.6~1.2 nm [18,19,20]. After stirring and sonication in the solution state, the GO was exfoliated into individual GO sheets. Then, the purified nonwoven fabric was impregnated with GO solution to make GO uniformly dispersed within the polypropylene nonwoven fabric. Finally, the GO in the nonwoven fabric was reduced to water-insoluble reduced graphene oxide (RGO), and the RGO/NW composite was obtained after washing and drying. Four 180 × 180 RGO/NW composites can be prepared in one preparation process, which has a lower filling rate, a higher yield, and a lower production cost compared to those of magnetic materials [13,21]. Four graphene samples of RGO/NWs were prepared by adjusting the GO concentration in the GO solution to 3, 5, 7, and 9 mg/mL, respectively. (The detailed preparation process is described in the supporting material.)

### 2.2. RGO/NW Composite Stability Experiments

Figure 2 shows the process of RGO/NW composite performance stability experiments. The complex application environment of an electromagnetic wave-absorbing wearable device is simulated, and several treatments, such as heavy extrusion, folding, immersion, high temperature, and low temperature are applied to test the stability of the dielectric and wave absorbing properties of the RGO/NW material. In this paper, the dielectric and absorbing properties of sample 3 were tested after 5, 10, 15, and 20 cycles of treatment, respectively. The specific experimental procedure is shown in Figure 2. First, sample 3 was pressed under a heavy weight for 10 h, removed and folded 50 times crosswise and longitudinally, then immersed in cold water for 2 h, and finally, held a constant temperature drying chamber at 300 °C for 4 h. After drying the sample, high-temperature experiments were conducted, and the sample was then put into a low-temperature test chamber at −50 °C for 2 h. A complete process was performed for 1 treatment, and after completing the 5th, 10th, 15th, and 20th treatments, the dielectric and absorbing properties were recorded. After the 5th, 10th, 15th, and 20th treatments, the thickness of the sample was recorded, and the dielectric and absorbing properties of the sample were tested.

### 2.3. Preparation of Multilayer RGO/NW Composites

The RGO/NW composites prepared from GO solutions at concentrations of 3, 5, 7, and 9 mg/mL were numbered 1, 2, 3, and 4. Different numbered single-layer RGO/NW composites were bonded according to the Jaumann absorber design results. Two, three and four layers of multilayer RGO/NW composites were prepared, respectively.

### 2.4. Characterization

The microscopic morphology of the RGO/NW material samples was characterized using a field emission scanning electron microscope (Hitachi S-4800, Tokyo, Japan). Based on the waveguide method, a vector network analyzer (Agilent E8363B, Santa Clara, CA, USA) was used to measure the dielectric constant of the RGO/NW composite samples. The arch method was used to measure the electromagnetic reflection loss (RL) of the RGO/NW materials in the range of 2–18 GHz.

## 3. Results and Discussion

### 3.1. Morphological Analyses

The four prepared single-layer RGO/NW composites (Figure 1) were characterized microscopically, and their SEM results were obtained, as shown in Figure 3. In Figure 3a, the graphene appears as a highly translucent and soft film, indicating that the Hummers’ method successfully exfoliated the multilayer GO flakes and reduced them to form an RGO with limited layers and flakes. Moreover, in addition to the larger graphene flake layers, many graphene fragments can be seen attached to the nonwoven fibers, but it is still too dispersed to form an effective conductive network. Figure 3b shows the 5 mg/mL GO solution prepared sample. Compared to Figure 3a, the concentration of graphene increases significantly, but it can only be present on the nonwoven fibers by attachment. As the concentration increased to 7 mg/mL, it was found (Figure 3c) that graphene began to fill the gaps between the nonwoven fibers, connecting the graphene attached to the fibers and forming a 3D graphene structure. The connection between graphene can form an extensive conduction network, while the resulting 3D graphene structure will contribute to the graphene “confinement effect” on the electromagnetic waves [22,23,24]. When the GO concentration increased to 9 mg/mL (Figure 3d), the dispersion of graphene became less effective, leading to the formation of dense graphene films in the nonwoven fiber space that enhanced the metallicity of RGO/NW, improving the electrical conductivity of the composite, and reducing the skinning depth of the composite. At a deeper level, such a high concentration of GO forms graphene with significantly more layers and smaller gaps, which would lead to a lower amount of energy required for the electron leap between graphene layers and the easy drift during charge transfer, resulting in more difficult regulation of graphene conductivity. Subject to the combined effect of the above two factors, the electromagnetic wave conduction loss capacity of the 9 mg/mL sample decreases, and the impedance matching degree with air decreases, resulting in a significant weakening of the wave absorption performance of the composite material.

### 3.2. Dielectric Performance Analysis of RGO/NW Composites

According to the Maxwell electromagnetic theory, the dielectric constant ($\varepsilon =\varepsilon^{\prime }+i\varepsilon^{\prime \prime }$) is often used to describe the reaction of a medium in an external electric field. The magnitude of the dielectric constant $\varepsilon^{\prime }$ directly affects the wavelength and refractive index of the electromagnetic wave incident inside the material, and $\varepsilon^{\prime \prime }$ indicates the ability of the electric dipole to dissipate energy as the frequency of the electromagnetic wave changes. Figure 4 shows the variation in the electromagnetic parameters of the graphene composites with GO concentration and frequency. Figure 4a,b shows that the values of $\varepsilon^{\prime }$ and $\varepsilon^{\prime \prime }$ are directly related to the concentration of GO used in the preparation process. The higher the concentration of GO, the higher the dielectric constant value. Further, when the GO concentration was increased from 5 to 7 mg/mL, there was a large gradient boost in both $\varepsilon^{\prime }$ and $\varepsilon^{\prime \prime }$ at 2.0 GHz from 5.3 to 7.2. This phenomenon may be caused by the fact that the concentration reaches the percolation threshold and that a 3D graphene structure is formed. The minimum filling rate to form a conductive network is usually referred to as the conductive percolation threshold [25]. Once the percolation threshold is exceeded, free electrons from the composite can overcome the resistivity and pass throughout the conduction network, resulting in a large increase in the dielectric constant. Under this effect, the difference in the refractive index of the electromagnetic waves between air and graphene becomes larger, forming a refractive boundary. This significantly enhances the field strength inside 3D graphene compared to the incident field, when the incident wave enters the 3D graphene structure. Such a local field enhancement is known as the “confinement effect and would further enhance the material’s dielectric constant. In addition, graphene prepared by the Hummers’ method forms a large number of defects and functional groups under a strong acid oxidation environment, which are prone to become polarization centers for electric dipole polarization under electromagnetic fields [26,27]. However, as the electromagnetic wave frequency increases, the dipole gradually fails to keep up with the change in frequency, causing a hysteresis phenomenon and eventually reaching the limit. For this reason, the contribution of the polarization relaxation to the dielectric constant decreases, and $\varepsilon^{\prime }$, $\varepsilon^{\prime \prime }$ gradually decrease with increasing frequency.

The dielectric loss tangent ($\tan\delta_{\varepsilon }=\varepsilon^{\prime \prime }/\varepsilon^{\prime }$) is generally used to evaluate the dielectric loss capacity of absorbing materials to electromagnetic waves, and the larger the value, the stronger the dielectric loss capacity. From the SEM, it can be known that the concentration of RGO in RGO/NW is positively correlated with the GO concentration, and the larger the GO concentration used for preparation under the same conditions, the larger the graphene concentration. As shown in Figure 4c, the low dielectric loss tangent values of samples of 3 mg/mL and 5 mg/mL imply poor electromagnetic wave loss capability. When the GO concentration was increased from 5 mg/mL to 7 mg/mL, there was a qualitative increase in the dielectric loss capacity of the samples due to the influence of the percolation threshold and the 3D graphene structure. For 7 mg/mL and 9 mg/mL samples, they both follow the law of concentration-determined dielectric constant, but the difference has become significantly smaller, and even at 10 GHz, the value of the dielectric loss constant of the 7 mg/mL sample exceeds that of the 9 mg/mL sample.

On the other hand, the degree of fluctuation of the loss tangent curve represents the degree of resonance between the graphene absorber and the electromagnetic wave for this sample. The resonance intensity increases as the concentration of graphene in RGO/NW increases. The tangent curve shows that the resonance between graphene and electromagnetic waves was not significantly enhanced when the GO concentration increased from 7 mg/mL to 9 mg/mL.

Figure 4d shows the conductivity of the four RGO/NW samples under different pressures. The conductivity of the dielectric materials can be expressed by Equation (1).
(1)$$\varepsilon =\varepsilon^{\prime }+i\frac{\sigma }{\omega }$$
where $\varepsilon^{\prime }$ is the real part of the dielectric constant, σ is the dielectric conductivity, and ω is the angular frequency of the electromagnetic wave. Therefore, the conductivity σ is positively correlated with $\varepsilon^{\prime \prime }$, the imaginary part of the dielectric constant. As shown in Figure 4d, the imaginary parts of the conductivity and the dielectric constant have similar basic laws, which are directly determined by the graphene concentration within the RGO/NW composite. The larger the concentration, the stronger the conductivity. Similarly, the increase in external pressure makes the space inside the nonwoven fabric smaller, which increases the concentration of graphene inside the RGO/NW and increases the chance and area of contact between the nonwoven fabric and graphene, leading to the increase in electrical conductivity with the increase in pressure. In addition, unlike the 9 mg/mL sample, the conductivity of the 7 mg/mL sample increased at an accelerated rate with pressure, and the conductivity of the two was almost equal when the pressure was 10 MPa. Of course, the conductivity of RGO/NW cannot grow indefinitely with the increase in concentration, and it will stop growing when it reaches a peak. In conclusion, when RGO/NW is used as a structural absorbing material, the pressure will greatly affect the conductivity of the material, and the high conductivity will enhance the skin effect and reduce the impedance matching.

### 3.3. Absorption Properties of RGO/NW Composites

#### 3.3.1. Analysis of Absorption Performance of Single-Layer RGO/NW Composites

The electromagnetic wave reflection loss (RL) of RGO/NW material in the range of 2~18 GHz was measured by the arch method, shown in Figure 5a. When the GO concentrations were 3 mg/mL and 5 mg/mL, the maximum RL intensity of RGO/NW was less than 10 dB, indicating that the single-layer RGO/NW prepared at GO concentrations lower than 5 mg/mL could not complete the effective absorption of electromagnetic waves. As mentioned in the dielectric property analysis, when the GO concentration reaches 7 mg/mL, the RGO/NW material reaches the percolation threshold, and the absorption performance is greatly enhanced. At this point, the effective absorption bandwidth of the composite material reaches 7.3 GHz (4.7–12 GHz), with a peak RL of −19.7 dB. When the concentration increases to 9 mg/mL, the absorption bandwidth (RL<−10 dB) of the RGO/NW material becomes 6.9 GHz, with a maximum reflection loss of −17.2 dB, which is slightly lower than that of the 7 mg/mL RGO/NW material. Combined with the microscopic morphology and dielectric property analysis, it is clear that the main factor affecting the absorption performance of this sample is the impedance matching between the composite and air.

In addition, the peak position of the curve shows that the wave peak of RL shifts to lower frequencies as the graphene concentration increases. Generally, an aluminum plate is used as the bottom reflector in the RL test process. According to the quarter-wavelength cancellation theory (Equation (2)), when the phase difference between the incident wave on the surface of the RGO/NW material and the reflected wave from the aluminum plate is one-quarter period, the two will produce interference phase extinction [21,28], as shown in Figure 5b.
(2)fpeak=(2n−1)c4d(Re[εrμr])
where $f_{peak}$ is the frequency of the wave peak appearance, c refers to the speed of light in a vacuum, *d* is the thickness of the absorber, *n* is the number of absorption peaks (*n* = 1,2,3 ……), and Re[εrμr] denotes the refraction coefficient within the absorber medium. As shown in Figure 5a, the absorption peaks of the four concentrations of the RGO/NW material samples appear at 10.8 GHz, 9.3 GHz, 7.3 GHz, and 5.9 GHz, respectively. It can be seen that the increase in graphene concentration in the composite material causes the increase in complex dielectric constant $\varepsilon_{r}$ and consequently, the refractive coefficient Re[εrμr], which leads to the decrease in absorption peak frequency $f_{peak}$. In summary, it can be determined that the absorption law of the single-layer RGO/NW composite is consistent with the quarter-wavelength cancellation theory. As a resonant structural absorbing material, the graphene concentration and the thickness of the nonwoven fabric are matched reasonably effectively.

#### 3.3.2. Structural and Electromagnetic Performance Stability of RGO/NW Composites

Figure 6 shows the samples’ thickness variation after different experiments. The initial thickness of sample 3 in the figure is 4.35 mm. The reason its thickness is lower than that of the original nonwoven fabric may be that the RGO/NW composite material needs to be sealed with cling film on the impregnated nonwoven fabric during the preparation process, and then put into a drying oven for heating reduction. During the drying process, the water inside the cling film-wrapped nonwoven fabric is evaporated into a large amount of vapor, and the high temperature and pressure formed by the vapor will compress the nonwoven fabric, resulting in a thinner thickness. After five treatments, the thickness increased by 0.4 mm to 4.75 mm, and after 10, 15, and 20 treatments, the thickness increased by 1.5 mm, 0.5 mm, and 0.5 mm, respectively. The sample thickness increased because the nonwoven fabric quickly recovered to its original thickness during the first five treatments. Therefore, even though the thickness of sample 3 of the graphene/nonwoven composite gradually increased with the number of treatments, it did not increase much. The thickness of sample 3 of the RGO/NW composite tends to be stable. It can be seen that the nonwoven fabric prepared from polypropylene fibers exhibits good deformation recovery ability and structural stability.

Figure 7a, b compares the dielectric constants of sample 3, before and after treatment. As can be seen from the figure, the real and imaginary parts of the dielectric constant of sample 3 decreased to a great extent after five treatments. Excluding the interference of measurement errors, the decrease in the dielectric constant between 10 and 20 treatments is very small.

Therefore, it can be concluded that the real and imaginary parts of the dielectric constant of the sample stabilized after five treatments. Furthermore, as can be seen in Figure 7c, the absorption bandwidth of the sample becomes progressively narrower after repeated treatments, while the absorption peak shifts to the right. In line with the variation of the dielectric constant, the absorption performance decreases significantly in the first measurement after 5 treatments, but stabilizes as the number increases. From the thickness change of sample 3, we can determine that there is a large increase in thickness after 5 treatments, and the space inside the nonwoven fabric increases and the concentration of graphene decreases, which leads to the decrease in dielectric and wave absorption properties. However, the RGO/NW composite shows quite stable dielectric and wave absorption properties with increased treatment times.

The analysis of the RGO/NW composites with good dielectric properties and electromagnetic wave absorption performance stability may be due to the following reasons.

The nonwoven fabric has good deformation recovery ability. Multiple foldings and pressings do not cause permanent deformation, thus ensuring a good spatial distribution of graphene inside the nonwoven fabric matrix.Graphene, as a carbon material, exhibits extremely stable physicochemical properties and does not react with oxygen or water in the air at 300 °C or −50 °C.The carbon atoms in graphene are connected by strong covalent bonds. When subjected to external mechanical forces, the carbon atoms are bent, but remain in a strict order, ensuring structural stability.Graphene is the material with the largest specific surface area found so far, and due to its light and flexible web-like structure, it is entangled between the fibers inside the nonwoven fabric and does not easily loosen from the nonwoven fabric matrix.Unlike graphene oxide prepared by the Hummers’ method, the chemically reduced graphene is not soluble in water, so it does not dissolve and disperse during immersion.

#### 3.3.3. Analysis of Absorption Performance of Multilayer HC Composite

It can be seen that the RGO/NW composite has good environmental adaptability and can maintain stable structure and wave absorption performance under various extreme environments, which meets the performance requirements of electromagnetic wave-absorbing wearable devices, showing high application value and prospects. RGO/NW composite is a typical resonant structure absorbing material, but limited by its thickness and its own electromagnetic properties, it is not sufficient to achieve broadband absorption. Therefore, according to the design concept of the *Jaumann* absorber, the multilayer structure is designed by matching RGO/NW composites with different dielectric properties based on the functional properties of the matching layer, absorber layer, and reflector layer.

In order to enhance its wave absorption performance and maintain the advantage of stable RGO/NW performance, four samples are now numbered 1, 2, 3, and 4, from low to high concentrations, in a multilayer design, as detailed in Table 1.

The RL of the multilayer RGO/NW in (Table 1) was measured by the arch method, as shown in Figure 8. The reflection loss curves of the double-layer RGO/NW samples in Figure 8a, with sample 1 as the upper matching layer and 2, 3, and 4 as the lower layer, are similar to those of the single-layer, with the wave peaks of the reflection loss curves shifting to lower frequencies as the graphene concentration of the samples increases. For single-layer samples 2, 3, and 4, the double-layer material formed with matching layer 1 broadens the absorption bandwidth, to some extent. However, the dielectric properties of sample 1 are too low, resulting in an insignificant frequency broadening effect. Moreover, as can be seen in Figure 8b, when 2 is used as the matching layer, the maximum absorption peak of 2-3 reaches −27.6 dB, and the effective absorption bandwidth is broadened to 8.6 GHz (3.8–12.1 GHz, 17.7–18 GHz). When combining 2 and 4, the absorption bandwidth reaches 11.3 GHz, and the unique double peaks are located at 5.6 GHz and 15.8 GHz, respectively, with an approximately 3-fold relationship, fully compounding the description of the phase extinction theory of the quadrature wavelength of λ/4 and 3λ/4. Figure 8c shows the reflection loss curves of the double-layer samples 3-4, from which it can be seen that the curves have only one wave peak, but the maximum reflection loss reaches −49.5 dB. This is because the single-layer sample 3 has higher dielectric properties and better absorption performance than samples 1 and 2, but at the same time, the reduced impedance matching performance determines that it is not suitable as a matching layer. The electromagnetic waves that can enter the inside of sample 3 suffer repeated losses in the resonant cavity between samples 3 and 4. Therefore, for the double-layer RGO/NW materials 3-4, the lack of a matching layer with good wave transmission does not improve the broadening the absorption bandwidth, but greatly enhances the absorption peak of 3-4 due to the formation of the resonant cavity.

Multilayer RGO/NW materials designed strictly according to the functional requirements of the matching, absorbing, and reflecting layers are divided into four types of three layers and one type of four layers. Figure 9 shows the RL at 2–18 GHz for the multilayer RGO/NW material. Each curve for the multilayer RGO/NW material in the figure has multiple peaks. It should be noted that the number of RL peaks for sample 1 as a matching layer is one less than the number of layers, i.e., two peaks for the three-layer material and three peaks for the four-layer material. The reason for this phenomenon is that the dielectric properties of sample 1 are too low to complete effective wave absorption alone.

The trends of the reflection loss curves of samples 2-3-4 show that the curves exhibit only one peak in the frequency range of 2–18 GHz, but it can be speculated that there are other peaks outside the frequency range of 2–18 GHz, which is of great interest for further studies. Among all the multilayer nonwoven samples, samples 1-2-3 showed the worst absorption performance, with an effective absorption bandwidth of 9.8 GHz (4.7–14.2 GHz, 17.7–18 GHz). It has a slightly wider absorption bandwidth than the double layer samples 2-3, but with reduced absorption intensity. Among the five samples, samples 1-3-4 showed a maximum absorption peak of −26.6 dB, yet could not achieve the full band absorption from 2 to 18 GHz. Besides, the remaining samples with three layers 1-2-4, 2-3-4, and four layers 1-2-3-4 have good performance and can accomplish full frequency absorption in the range of 2–18 GHz. Based on the above phenomena, it can be seen that the matching layer, absorbing layer, and reflecting layer must be reasonably matched. When using sample 1 as the matching layer of a multilayer RGO/NW composite, the best lower absorber layer is sample 2. If 3 or 4 is used as the absorber layer, the absorption effect of the multilayer sample is necessarily poor. Therefore, the three-layer 1-2-4, 2-3-4 and four-layer 1-2-3-4 can be fully compatible with the design concept of a Jaumann absorber, with better absorption performance.

Moreover, the three-layer RGO/NW 1-2-4 and 2-3-4 accomplish 97% electromagnetic wave absorption (RL<−15 dB) at 3.2–13.8 GHz and 2–14.3 GHz, respectively. The four-layer 1-2-3-4 sample can achieve 97% absorption of nearly the full frequency band from 2 to 18 GHz, and even 99% EM wave absorption (RL<−20 dB) at the 3.8–14.8 GHz band. Such good absorption performance is attributed to the reasonable gradient dielectric design and the formation of resonant cavities between the dielectric layers. The lowest dielectric concentration is used as the matching layer of the multilayer material, so that most of the electromagnetic waves in contact with the absorbing material can enter the absorbing material. Then, it enters the absorbing and reflecting medium layer with higher concentration for multiple scattering and dielectric loss to dissipate the electromagnetic wave energy in the form of heat or mechanical energy.

The electromagnetic wave absorption mechanism of RGO/NW is shown in Figure 10. At the microscopic scale, the 3D graphene created by the non-woven fiber and graphene combined provides a good structural condition for the domain-limiting effect of graphene. The electromagnetic waves entering 3D graphene can be multiply scattered and repeatedly absorbed by the limiting field effect. The polarization relaxation and conduction losses are mainly responsible for the absorption of electromagnetic waves. At the macro scale, the multilayer RGO/NW composites designed by the *Jaumann* absorber can greatly broaden the effective absorption band and increase the absorption intensity. The sequential superposition of the matching layer, absorption layer, and reflection layer can improve the impedance matching of RGO/NW. The progressively increasing dielectric concentration between the layers improves the material’s overall impedance matching degree and electromagnetic wave attenuation ability. At the same time, the formed heterogeneous interface leads to multiple scattering between the layers, further promoting the absorption of electromagnetic waves. The main result is that as the number of layers increases, the frequency range in which the quarter-wavelength cancellation can function increases, directly affecting the absorption bandwidth of the RGO/NW composite.

## 4. Conclusions

A stable dual 3D structure was formed by dispersing graphene in the nonwoven matrix using Hummers’ method. The extreme environmental adaptation test proved that the prepared RGO/NW flexible composite has good structural recovery properties and stable wave absorption performance. By means of a multilayer structure design, the flexible RGO/NW composite obtained broadband absorption performance. The 2-3 samples of two-layer RGO/NW composites have the widest absorption bandwidth (<−10dB) of 8.6 GHz; 1-2-4 and 2-3-4 samples of three-layer RGO/NW composites both achieve full-band absorption (<−10dB) from 2 to 18 GHz; 1-2-3-4 samples of four-layer RGO/NW composites achieve reflection loss below −15 dB at 2 to 18 GHz and <−20 dB reflection loss from 3.8 to 14.8 GHz. Such multilayer graphene/nonwoven composites can form good electromagnetic wave protection on the surface of objects with complex structural shapes, including personnel wearable devices, and can perform well in a wide range of extreme environments.

## Figures and Tables

**Figure 1 nanomaterials-13-00634-f001:**
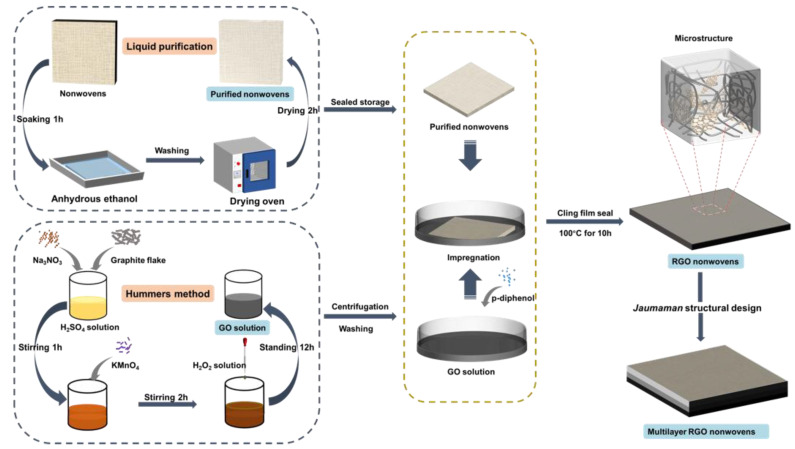
Preparation process of RGO/NW composite.

**Figure 2 nanomaterials-13-00634-f002:**
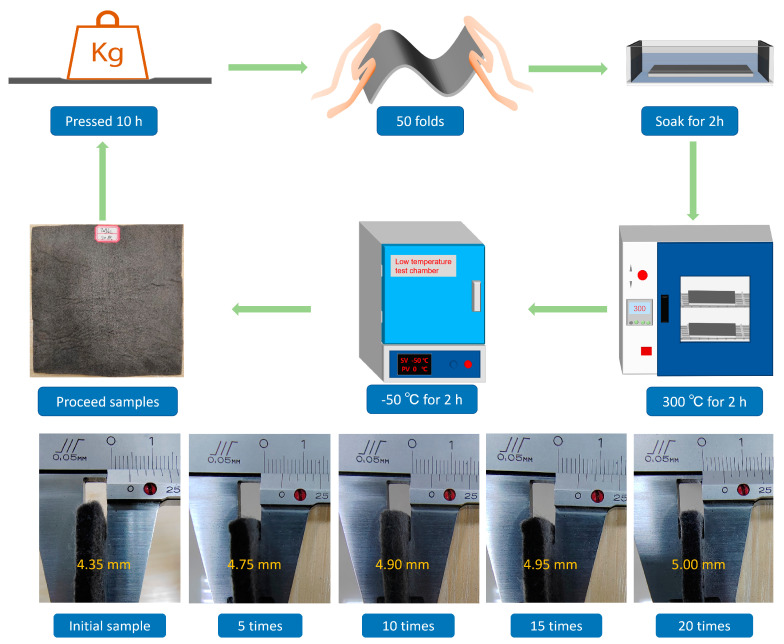
Extreme environmental adaptability test of RGO/NW composites.

**Figure 3 nanomaterials-13-00634-f003:**
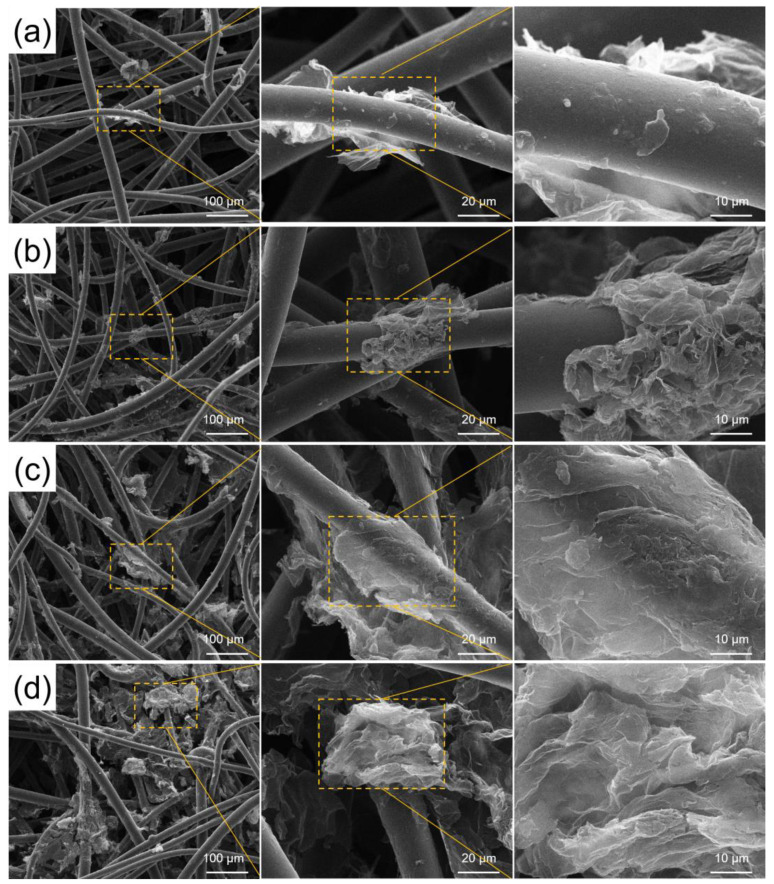
SEM of RGO/NW prepared by different GO solution concentrations: (**a**) 3 mg/mL; (**b**) 5 mg/mL; (**c**) 7 mg/mL; (**d**) 9 mg/mL.

**Figure 4 nanomaterials-13-00634-f004:**
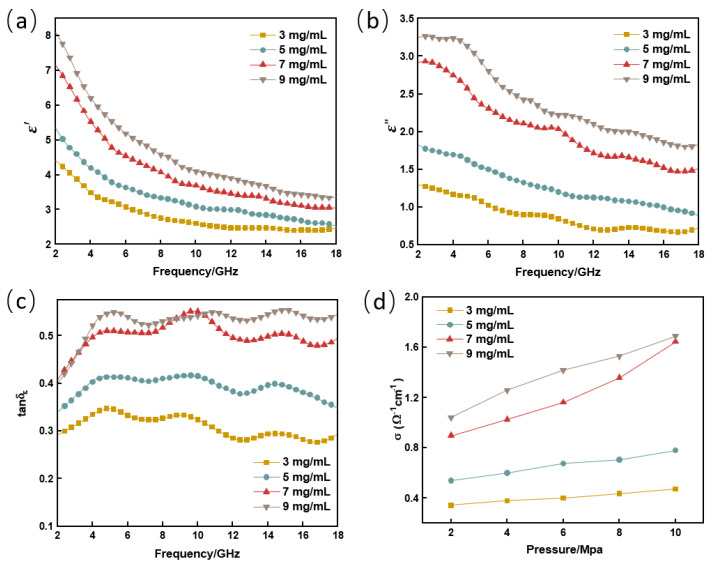
Electromagnetic parameters of the RGO/NW composite: (**a**) real part of the dielectric constant $\varepsilon^{\prime }$; (**b**) imaginary part of the dielectric constant $\varepsilon^{\prime \prime }$; (**c**) dielectric loss tangent value $\tan\delta_{\varepsilon }$; (**d**) electrical conductivity σ.

**Figure 5 nanomaterials-13-00634-f005:**
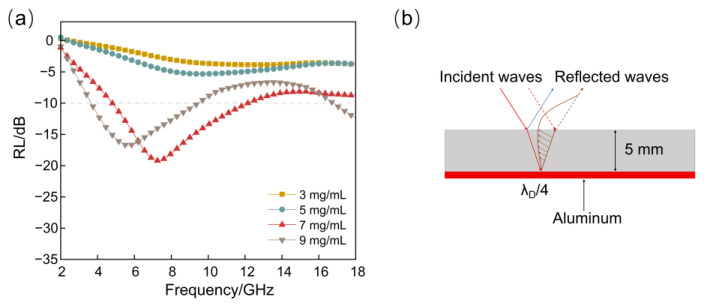
(**a**) Reflection loss of four concentration samples; (**b**) schematic diagram of quarter-wavelength phase extinction of single-layer samples.

**Figure 6 nanomaterials-13-00634-f006:**
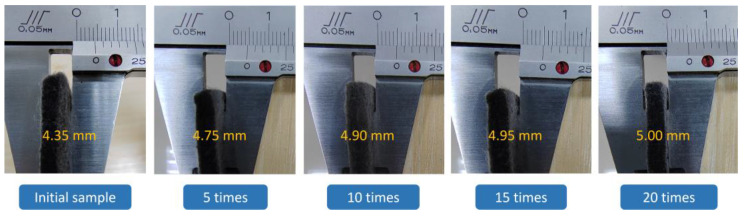
Thickness of samples after different numbers of treatments.

**Figure 7 nanomaterials-13-00634-f007:**
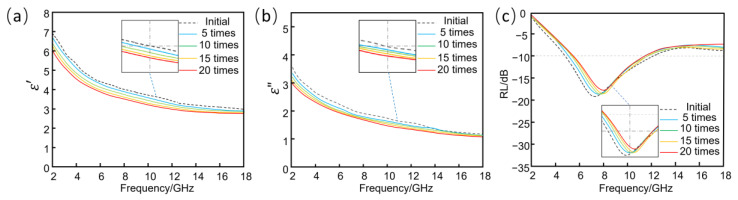
Electromagnetic properties after different treatment times: (**a**) real part of dielectric constant; (**b**) imaginary part of dielectric constant; (**c**) reflection loss.

**Figure 8 nanomaterials-13-00634-f008:**
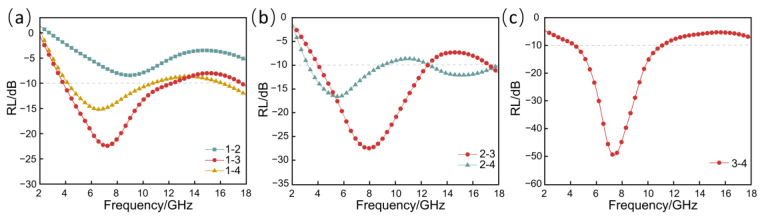
RL curves of double-layer RGO/NW. (**a**) 1-2, 1-3, 1-4; (**b**) 2-3, 2-4; (**c**) 3-4.

**Figure 9 nanomaterials-13-00634-f009:**
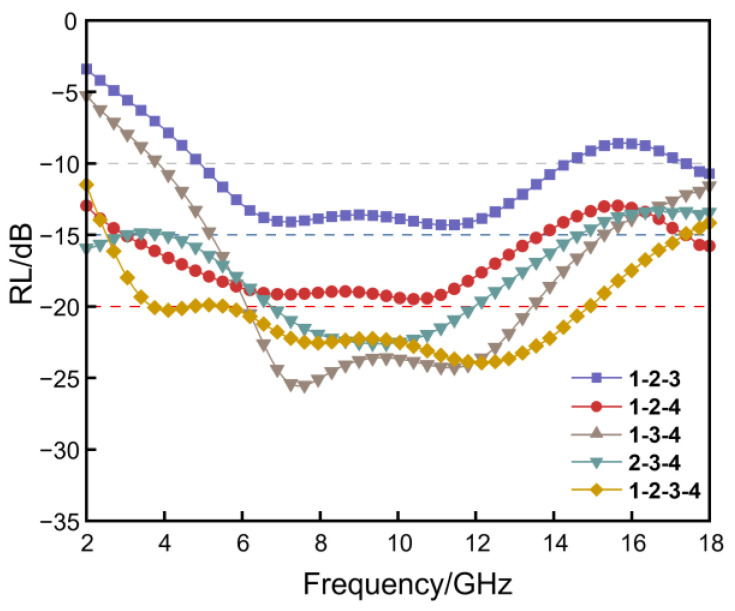
RL curves of multilayer RGO/NW composites.

**Figure 10 nanomaterials-13-00634-f010:**
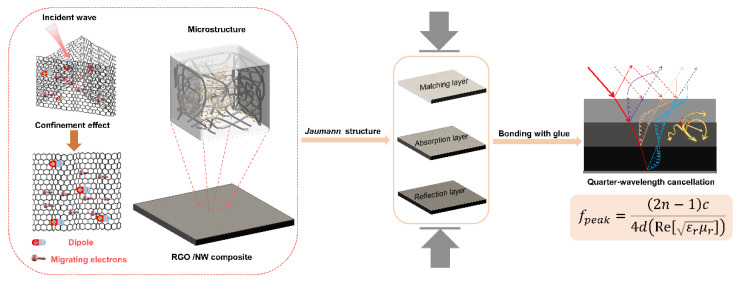
Schematic diagram of the wave absorption mechanism of RGO/NW.

**Table 1 nanomaterials-13-00634-t001:** Design options for multilayer RGO/NW.

Sample Code	Type	Thickness	Stacking Order (High to Low)
#1-2	Double-layer	10 mm	1, 2
#1-3		10 mm	1, 3
#1-4		10 mm	1, 4
#2-3		10 mm	2, 3
#2-4		10 mm	2, 4
#3-4		10 mm	3, 4
#1-2-3	Three-layer	15 mm	1, 2, 3
#1-2-4		15 mm	1, 2, 4
#1-3-4		15 mm	1, 3, 4
#2-3-4		15 mm	2, 3, 4
#1-2-3-4	Four-layer	20 mm	1, 2, 3, 4

## Data Availability

No new data were created.

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
