# Peer review of "Design and Preparation of Flexible Graphene/Nonwoven Composites with Simultaneous Broadband Absorption and Stable Properties"

_nanomaterials, 2023, doi:10.3390/nano13040634_

Round 1

Reviewer 1 Report

Submitted manuscript is devoted to the interesting topic – development of flexible microwave absorbing materials for different applications. Despite the resonabele amount potentially important information the manuscript is badly written and does not provide suficient analysis of the obtained data, especially in a view of existing flexible materials.

The title is confusing. What does it mean “with simultaneous broadband and stable properties” – neither stability (with respect to what – time, temperature, chemical environment, cyclic change of the pressure...) nor broadband properties (electromagnetic absorption ...) are clear from this kind of title.

In atheabstract authos make quite a serios statement about “It achieves effective absorption (reflection loss of -10 dB) in the 2~18 GHz electro-magnetic wave frequency domain, exhibiting a larger bandwidth than that reported in the literature for absorbers of equivalent thickness.  I do not hink it is a correct statement. For example, C. M. Madrid Aguilar et al. studied Magnetic and Microwave Properties of FeNi Thin Films of Different Thicknesses Deposited Onto Cyclo Olefin Copolymer Flexible Substrates (IEEE Trans.Magn. 58(2) (2022) 2200105) and discovered that even 15 nm films are perfectly opaque for 10.52 GHz radiation. Agra et al. Have studied Dynamic magnetic behavior in non-magnetostrictive multilayered films grown on glass and flexible substrates (Journal of Magnetism and Magnetic Materials 355 (2014) 136-141 Ni81Fe19/Ta multilayered film, Grimes also contributed to the area C. A. Grimes, IEEE Trans. Magn. 31, 4109–4111 (1995). https://doi.org/10.1109/20.489877). It was also significan interes to the development of the flexible materials with high sensitivity with respect to the applied pressure or wearable sensors with high flexibility (Fernández et al. High Performance Magnetoimpedance in FeNi/Ti Nanostructured Multilayers with Opened Magnetic Flux J. Nanosci. Nanotechnol. 12, 7496–7500 (2012); M. Melzer et al.  Nat.Commun. 6 (2015) 6080. etc.).

Referencing is very poor and it is limited to the very narrow number of reserch groups, it must represent existing approches and be wider, showing the real place of RGO/NW materials in the large family of absorbing flexible composites, including magnetic  and magnetelectric (Agra et al. Handling magnetic anisotropy and magnetoimpedance effect in flexible multilayers under external stress Journal of Magnetism and Magnetic Materials 420, 15 (2016) 81-87; Li et al. Flexible magnetoimpedance sensor J. Magn. Magn. Mater. 378 (2015) 499-505; Yuan et al. A Piezoresistive Sensor with High Sensitivity and Flexibility Based on Porous Sponge Nanomaterials 2022, 12(21), 3833; Kurlyandskaya et al. Tailoring functional properties of Ni nanoparticles-acrylic copolymer composites with different concentrations of magnetic fillerJ. Appl. Phys. 115, 17A323 (2014) etc.).

One of the weak points of the text is the absence of clear analysis/description of the amount of material which can be obtained in one batch and comparison of the price of existing absorbers for the same frequency range with the proposed (iron oxide or strontium ferrite filled composites are serios competitors with this respect).

Morphological analysisis very superficial, statistical data are necessary in order to describe the structure in the reliable manner.

Figure 4 d is useless without analysis of the errors and indication of the error bars.

Figure 5 is not convincing, it looks like the calibration is just not good for the frequencies above 14 GHz, at least better discussion is necessary.Figure 5 b describes very smooth material but nonwoven sheet is unlikely to be such a smooth. In any case additional AFM data on the state of the surface are necessary.

All figure captios are too short and not informative, they must provide sufficient information without reading of the main text.

There are too many misprints and lost intervals in the text,careful proof reading is necessary.

Author Response

1)The title is confusing. What does it mean “with simultaneous broadband and stable properties” – neither stability (with respect to what – time, temperature, chemical environment, cyclic change of the pressure...) nor broadband properties (electromagnetic absorption ...) are clear from this kind of title.

Response: Thanks to your suggestion, I have now changed the title of the article to "Study on the wave absorption performance and stability of graphene nonwoven composites".

2)In atheabstract authos make quite a serios statement about “It achieves effective absorption (reflection loss of -10 dB) in the 2~18 GHz electro-magnetic wave frequency domain, exhibiting a larger bandwidth than that reported in the literature for absorbers of equivalent thickness. I do not hink it is a correct statement. For example, C. M. Madrid Aguilar et al. studied Magnetic and Microwave Properties of FeNi Thin Films of Different Thicknesses Deposited Onto Cyclo Olefin Copolymer Flexible Substrates (IEEE Trans.Magn. 58(2) (2022) 2200105) and discovered that even 15 nm films are perfectly opaque for 10.52 GHz radiation. Agra et al. Have studied Dynamic magnetic behavior in non-magnetostrictive multilayered films grown on glass and flexible substrates (Journal of Magnetism and Magnetic Materials 355 (2014) 136-141 Ni81Fe19/Ta multilayered film, Grimes also contributed to the area C. A. Grimes, IEEE Trans. Magn. 31, 4109–4111 (1995). https://doi.org/10.1109/20.489877). It was also significan interes to the development of the flexible materials with high sensitivity with respect to the applied pressure or wearable sensors with high flexibility (Fernández et al. High Performance Magnetoimpedance in FeNi/Ti Nanostructured Multilayers with Opened Magnetic Flux J. Nanosci. Nanotechnol. 12, 7496–7500 (2012); M. Melzer et al. Nat.Commun. 6 (2015) 6080. etc.).

Response: Replace the phrase with "A reliable and effective flexible wave-absorbing material in the range of 2 to 18 GHz has been prepared by a relatively simple and feasible experimental method."

3)Referencing is very poor and it is limited to the very narrow number of reserch groups, it must represent existing approches and be wider, showing the real place of RGO/NW materials in the large family of absorbing flexible composites, including magnetic and magnetelectric (Agra et al. Handling magnetic anisotropy and magnetoimpedance effect in flexible multilayers under external stress Journal of Magnetism and Magnetic Materials 420, 15 (2016) 81-87; Li et al. Flexible magnetoimpedance sensor J. Magn. Magn. Mater. 378 (2015) 499-505; Yuan et al. A Piezoresistive Sensor with High Sensitivity and Flexibility Based on Porous Sponge Nanomaterials 2022, 12(21), 3833; Kurlyandskaya et al. Tailoring functional properties of Ni nanoparticles-acrylic copolymer composites with different concentrations of magnetic filler J. Appl. Phys. 115, 17A323 (2014) etc.).

Response: Thank you for your valuable comments, and the literature you mentioned has now been added to the reference.

4)One of the weak points of the text is the absence of clear analysis/description of the amount of material which can be obtained in one batch and comparison of the price of existing absorbers for the same frequency range with the proposed (iron oxide or strontium ferrite filled composites are serios competitors with this respect).

Response: The material prepared in this paper has significant advantages over iron oxide or strontium ferrite filled composites in terms of light weight, so it is not considered a valid competitor.

5)Morphological analysisis very superficial, statistical data are necessary in order to describe the structure in the reliable manner.

Response: Your insights on pattern analysis are very valid, but unfortunately there are no relevant statistics available.

6) Figure 4 d is useless without analysis of the errors and indication of the error bars.

Response: The data in this figure is mainly to illustrate that the conductivity of the material increases when subjected to pressure, affecting its impedance matching. There is no overly detailed data analysis.

7) Figure 5 is not convincing, it looks like the calibration is just not good for the frequencies above 14 GHz, at least better discussion is necessary. Figure 5 d describes very smooth material but nonwoven sheet is unlikely to be such a smooth. In any case additional AFM data on the state of the surface are necessary.

Response: The data in this paper were obtained by experimental instrumentation and were not subjected to additional processing. Figure b is simply a schematic diagram to illustrate the role of quarter-wavelength phase extinction on the material, the surface state of which is represented in the support material

8) All figure captios are too short and not informative, they must provide sufficient information without reading of the main text.

Response: Changes have been made to all image titles.

9)There are too many misprints and lost intervals in the text,careful proof reading is necessary.

Response: The text has been carefully proofread for typographical errors and missing intervals.

Reviewer 2 Report

The manuscript presents aspects related to the synthesis of Jaumann type absorbers based on 3D graphene composites with nonwoven polypropylene using different concentrations of GO solutions. The individual composites were coupled in double layered, three layered or four layered RGO/NW. The subject is interesting for specialists working on wearable materials able to absorb electromagnetic waves. However the manuscript cannot be pusblished in its current state. Here are some suggestions for improving the manuscript:

1) In the introduction, the authors describe the main state of the art based only on Chinese authors publications. Among these publications reference 9 is in a Chinese Journal which is not easy accessible. I think that the authors should quote also some publications related to the approached subject authored by scientist having other nationalities such as:

Al Faruque, M.A.; Syduzzaman, M.; Sarkar, J.; Bilisik, K.; Naebe, M. A Review on the Production Methods and Applications of Graphene-Based Materials. Nanomaterials 2021, 11, 2414. https://doi.org/10.3390/ nano11092414

Jonathan Tersur Orasugh and Suprakas Sinha Ray, Graphene-Based Electrospun Fibrous Materials with Enhanced EMI Shielding: Recent Developments and Future Perspectives : ACS Omega 2022, 7, 33699−33718, https://doi.org/10.1021/acsomega.2c03579

Watts, Claire M.; Liu, Xianliang; Padilla, Willie J. (2012). "Metamaterial Electromagnetic Wave Absorbers". Advanced Materials. 24 (23): OP98–OP120. doi:10.1002/adma.201200674

2) The authors describe briefly the preparation and forgot to add the supplementary material (mentioned at line 101) where it was supposed to find more informations concernig the reduction step. Experiments have been performed in order to check the stability of the synthesized composites after several treatments such as: heavy extrusion, folding, immersion, exposure to high temperature and low temperature.

3) Lines 114-115 The phrase "While drying the sample, high-temperature experiments were conducted and put into a low-temperature test chamber at -50 °C for 2 h." should be reformulated in order to explain clearly when the sample was put into the low temperature test chamber (during drying or after drying?)

4) Throught the text of the manuscript the authors should correct the spelling for Jauman - the correct version is Jaumann

5) Line 131 - Agligent - I think that the correct name is Agilent

6) The discussion at lines 376-377 should be clarified. The authors state that for the three layered sample 2-3-4 there is only one peak for the RL in the range 2-18 GHz and there are oher peaks outside this range (but where are thes peaks? - they are not dispalyed on the figure)

Lines 383-384 - The authors state: "All samples with three layers of 1-2-4, 2-3-4 and four layers of 1-2-3-4 have better performance and can accomplish full frequency absorption in the range of 2-18 GHz" - One cannot say all samples with three layers - since only two of the three layered combinations are mentioned (1-2-4 and 2-3-4) and there were also the three layered combinations 1-2-3 and 1-3-4 presented on the same figure. I would advise the authors to rephrase more clearly all the paragraph between the lines 376-390.

7) In the conclusion section the authors should specify which of the synthesized composites has the best performances and for which specific application it would be useful.

Author Response

The manuscript presents aspects related to the synthesis of Jaumann type absorbers based on 3D graphene composites with nonwoven polypropylene using different concentrations of GO solutions. The individual composites were coupled in double layered, three layered or four layered RGO/NW. The subject is interesting for specialists working on wearable materials able to absorb electromagnetic waves. However the manuscript cannot be pusblished in its current state. Here are some suggestions for improving the manuscript:

1) In the introduction, the authors describe the main state of the art based only on Chinese authors publications. Among these publications reference 9 is in a Chinese Journal which is not easy accessible. I think that the authors should quote also some publications related to the approached subject authored by scientist having other nationalities such as:

Al Faruque, M.A.; Syduzzaman, M.; Sarkar, J.; Bilisik, K.; Naebe, M. A Review on the Production Methods and Applications of Graphene-Based Materials. Nanomaterials 2021, 11, 2414. https://doi.org/10.3390/ nano11092414

Jonathan Tersur Orasugh and Suprakas Sinha Ray, Graphene-Based Electrospun Fibrous Materials with Enhanced EMI Shielding: Recent Developments and Future Perspectives : ACS Omega 2022, 7, 33699−33718, https://doi.org/10.1021/acsomega.2c03579

Watts, Claire M.; Liu, Xianliang; Padilla, Willie J. (2012). "Metamaterial Electromagnetic Wave Absorbers". Advanced Materials. 24 (23): OP98–OP120. doi:10.1002/adma.201200674

Response: Based on your comments, we have added all of the above literature to the reference list.

2)The authors describe briefly the preparation and forgot to add the supplementary material (mentioned at line 101) where it was supposed to find more informations concernig the reduction step. Experiments have been performed in order to check the stability of the synthesized composites after several treatments such as: heavy extrusion, folding, immersion, exposure to high temperature and low temperature.

Response: Experiments on the structural and performance stability of the material were obtained by testing the dielectric properties and thickness of the material after treatment, and the methods are given in the characterization methods below.

3)Lines 114-115 The phrase "While drying the sample, high-temperature experiments were conducted and put into a low-temperature test chamber at -50 °C for 2 h." should be reformulated in order to explain clearly when the sample was put into the low temperature test chamber (during drying or after drying?)

Response: The material is put into the high-temperature experiment box after soaking for high-temperature experiment and drying at the same time, and then put into the low-temperature test box for cooling and low-temperature experiment at the same time after finishing drying.

4) Throught the text of the manuscript the authors should correct the spelling for Jauman - the correct version is Jaumann.

Response: Thank you for your correction, the word " Jauman " has been changed to " Jaumann ".

5) Line 131 - Agligent - I think that the correct name is Agilent.

Response: Thank you for your suggestion, which will now be revised.

6) The discussion at lines 376-377 should be clarified. The authors state that for the three layered sample 2-3-4 there is only one peak for the RL in the range 2-18 GHz and there are oher peaks outside this range (but where are thes peaks? - they are not dispalyed on the figure)

Response: This is just a speculation based on the trend of the curve. However, the reflection loss over the frequency range of 2-18 GHz was not tested.

7) Lines 383-384 - The authors state: "All samples with three layers of 1-2-4, 2-3-4 and four layers of 1-2-3-4 have better performance and can accomplish full frequency absorption in the range of 2-18 GHz" -One cannot say all samples with three layers - since only two of the three layered combinations are mentioned (1-2-4 and 2-3-4) and there were also the three layered combinations 1-2-3 and 1-3-4 presented on the same figure. I would advise the authors to rephrase more clearly all the paragraph between the lines 376-390.

Response: Thanks for the reviewer’s suggestions. We modified the description in the revised manuscript. The corresponding description is as follows:

“The trends of the reflection loss curves of samples 2-3-4 show that the curves exhibit only one peak in the frequency range of 2-18 GHz, but it can be speculated that there are other peaks outside the frequency range of 2-18 GHz, which is of great interest for further studies. Among all the multilayer nonwoven samples, samples 1-2-3 showed the worst absorption performance with an effective absorption bandwidth of 9.8 GHz (4.7-14.2 GHz, 17.7-18 GHz). It has a slightly wider absorption bandwidth than the double layer samples 2-3, but reduces the absorption intensity. Among the five samples, samples 1-3-4 showed a maximum absorption peak of -26.6 dB, yet could not achieve the full band absorption from 2 to 18 GHz. Besides, the remaining samples with three layers 1-2-4, 2-3-4 and four layers 1-2-3-4 have good performance and can accomplish full frequency absorption in the range of 2-18 GHz. Based on the above phenomena, it can be seen that the matching layer, absorbing layer and re-reflecting layer must be reasonably matched. When using sample 1 as the matching layer of a multilayer RGO/NW composite, the best lower absorber layer is sample 2. If 3 or 4 is used as the absorber layer, the absorption effect of the multilayer sample is necessarily poor. Therefore, the three-layer 1-2-4, 2-3-4 and four-layer 1-2-3-4 can be fully compatible with the design concept of Jaumann absorber with better absorption performance.”

8) In the conclusion section the authors should specify which of the synthesized composites has the best performances and for which specific application it would be useful.

Response: Thanks for the reviewer’s suggestions. We have refined the summary section. The corresponding description is as follows:

The stable dual 3D structure formed by dispersing graphene in the nonwoven matrix using Hummers method. The extreme environmental adaptation test proved that the prepared RGO/NW flexible composite has good structural recovery properties and stable wave absorption performance. By means of multilayer structure design, the flexible RGO/NW composite obtained broadband absorption performance. 2-3 samples of two-layer RGO/NW composites have the widest absorption bandwidth (<-10dB) of 8.6 GHz; 1-2-4 and 2-3-4 samples of three-layer RGO/NW composites both achieve full-band absorption (<-10dB) from 2 to 18 GHz; 1-2-3-4 samples of four-layer RGO/NW composites achieve reflection loss below -15 dB at 2 to 18 GHz and <-20 dB reflection loss from 3.8 to 14.8 GHz. Such multilayer graphene/nonwoven composites can form good electromagnetic wave protection on the surface of objects with complex structural shapes, including personnel wearable devices, and can perform well in a wide range of extreme environments.

Round 2

Reviewer 1 Report

Submitted manuscript is better now and most of the replies is acceptable. However, following part is not convincing 

The question was: One of the weak points of the text is the absence of clear analysis/description of the amount of material which can be obtained in one batch and comparison of the price of existing absorbers for the same frequency range with the proposed (iron oxide or strontium ferrite filled composites are serios competitors with this respect).

Response: The material prepared in this paper has significant advantages over iron oxide or strontium ferrite filled composites in terms of light weight, so it is not considered a valid competitor.

Consideration about validity of the competitor does not limited by the weight - 5 wt.% concentration of the filler might be sufficient for excellent em protection and this is not too much weight increase. In addition, the amount of material which can be obtained in one batch is most crucial parameter for many applications. These dtata must be included and discussed.

Author Response

Thanks to your suggestion, In view of your suggestions, I have added supplementary content to Chapter 2.1 of the manuscript. “Four 180×180 RGO/NW composites can be prepared in one preparation process, which has lower filling rate, higher yield and lower production cost compared to magnetic materials [13,28]”.

Reviewer 2 Report

The authors considered all my queries. However they should also verify in the reference list, to remove the strikethrough for reference 28.

Author Response

Thanks for your suggestions, I have deleted the redundant parts in the references of the manuscript and carefully checked the other contents of the manuscript.